# Tooth Diversity Underpins Future Biomimetic Replications

**DOI:** 10.3390/biomimetics8010042

**Published:** 2023-01-18

**Authors:** Di Wang, Shuangxia Han, Ming Yang

**Affiliations:** State Key Laboratory of Inorganic Synthesis and Preparative Chemistry, Jilin University, Changchun 130012, China

**Keywords:** tooth diversity, hierarchical structure, biomimetic materials, mechanical properties, multifunctionalities

## Abstract

Although the evolution of tooth structure seems highly conserved, remarkable diversity exists among species due to different living environments and survival requirements. Along with the conservation, this diversity of evolution allows for the optimized structures and functions of teeth under various service conditions, providing valuable resources for the rational design of biomimetic materials. In this review, we survey the current knowledge about teeth from representative mammals and aquatic animals, including human teeth, herbivore and carnivore teeth, shark teeth, calcite teeth in sea urchins, magnetite teeth in chitons, and transparent teeth in dragonfish, to name a few. The highlight of tooth diversity in terms of compositions, structures, properties, and functions may stimulate further efforts in the synthesis of tooth-inspired materials with enhanced mechanical performance and broader property sets. The state-of-the-art syntheses of enamel mimetics and their properties are briefly covered. We envision that future development in this field will need to take the advantage of both conservation and diversity of teeth. Our own view on the opportunities and key challenges in this pathway is presented with a focus on the hierarchical and gradient structures, multifunctional design, and precise and scalable synthesis.

## 1. Background

Paleontologists and anthropologists have long studied teeth because they degrade much more slowly than other biological tissues [1,2,3,4]. In fact, teeth, especially those comprising enamel, are the source of much of what we know about many ancient species [5,6]. These facts are consistent with the high content of minerals that can be preserved as fossils [7,8]. Despite being the hardest tissue with minor organic inclusions, tooth enamel is resilient and damage tolerant and is important for fulfilling the main functions of mastication, cutting, and grinding [9,10]. The exceptional mechanical properties of tooth enamel are a result of convergent evolution that selects the prevailing fibrous minerals [10,11,12,13,14,15]. This key anisotropic element endows tooth enamel with high modulus and hardness [16,17]. The usually minor proteins are, however, critical for energy dissipation within the multiple hierarchies [18,19].

The highly conserved tooth structure accompanies the remarkable diversity in different species [20,21]. The underlying principle for conservation and diversity of teeth is both ascribed to natural evolution for the targeted functions. For example, whereas most mammals are diphyodont—that is, they replace teeth only twice—most other vertebrates are polyphyodont, continuously replacing worn teeth throughout the lifetime [22,23,24]. This difference in the regeneration ability may indicate the different natural design criteria for human teeth and shark teeth. Human teeth have more complex structure to ensure the durability and minimize fracture vulnerability [25]. On the other hand, the high bite forces are more important for the sharp shark teeth to break into preys [26]. Similarly, although the calcite teeth in sea urchin are more brittle than human teeth, they are self-sharpening and can continuously grow [27]. Other rock scrapers such as chiton, maximize the hardness by choosing iron oxide as the biominerals [28,29,30]. Among mammals, different feeding habits of herbivores and carnivores alter the ultrastructure of their teeth to select for their most critical functions, such as anti-wear, antifatigue, and anti-fracture [24,31,32]. Knowledge on tooth diversity is, therefore, as important as their conservation in understanding and imitating the optimized hypermineralized structures for desirable applications.

Common in natural evolution is also its elegant solution to combine conflicting properties, which has fascinated scientists with regard to implementing similar design principles to boost material performance [33,34,35,36,37]. Although extensive studies have been performed to replicate the structures of nacre and bones [38,39,40,41,42], recent success in the biomimetic synthesis of enamel-like structures provides a new pathway for solving the potential technological problems and achieving demanding property sets. Although this field just emerged, these columnar nanocomposites already show exceptional combinations of mechanical properties [43,44,45], which are normally difficult to obtain using other design parameters. Although the current focus of tooth-inspired materials is on the highly conserved fibrous structures of enamel [43,44,45], the recognition of tooth diversity will provide more valuable inspirations for purpose-driven material design.

In this contribution, in a hope to push this emerging field further, we survey the current knowledge of teeth from representative mammals and aquatic animals, which can be used as potential biomimetic prototypes. Our focus is on tooth diversity with regard to aspects such as compositions, structures, properties, and functions. Different animal teeth are covered (Figure 1), ranging from well-studied human teeth, shark teeth, chiton teeth, and sea urchin teeth to the recently discovered transparent dragonfish teeth, stiffest black drum fish teeth, and many more. Special attention is paid to the studies that investigated the mechanical properties. The state-of-the-art of biomimetic materials inspired by enamel are briefly discussed, and the current limitations are highlighted. This review ends by providing our viewpoint of current challenges and future directions when using remarkably diverse teeth as prototypes for biomimetic replications.

## 2. Tooth Diversity

### 2.1. Mammal Teeth

#### 2.1.1. Human Teeth

##### Enamel

Enamel, the hardest material in the human body, consists of 96% hydroxyapatite (HAP) and 4% water and proteins, mainly amelogenin and enamelin. It constitutes the outermost layer of teeth with a thickness up to several mm [46], forming a barrier to protect teeth from external physical and chemical damage. Enamel has a hardness of 5.00 ± 0.22 GPa [47], a modulus of 97.12 ± 2.95 GPa [47], and a fracture toughness of 0.67 ± 0.12 MPa m^1/2^ [48], much higher than that of pure HAP of 0.3 MPa m^1/2^ [49]. The superior fracture tolerance of enamel is mainly attributed to its hierarchical structure [9,50]. The crystal first forms mineral nanofibers (30–40 nm), which gather into fiber bundles (80–130 nm) and then assemble into prisms (6–8 μm) [50]. Human enamel is characterized by the predominance of key-hole-shaped prisms among different Boyde’s patterns (Figure 2A) [51]. These prisms are assembled into strips, which are arranged differently across the thickness of enamel layer, defining enamel types. On the occlusal surface, prismless enamel may also exist [51]. In the outer enamel, the prisms are almost parallel to each other as radial enamel. However, in the inner enamel, prisms are cross arranged, forming decussating enamel. There are two characteristic areas in the inner enamel, namely the para-zone (P-zone) and the dia-zone (D-zone), according to the orientation of bands of prisms sectioned relative to their long axes [52], corresponding to the well-developed Hunter–Schreger bands [53,54]. Surrounding the prisms is interprismatic enamel. In interprismatic enamel, which is distinct from crystallites within prisms that are oriented roughly parallel to the prism long axis, the crystallites fan out from the center toward the edges. The boundaries between prismatic and interprismatic enamels appear as distinct structures as protein sheaths. At the next level, *schmelzmuster* comprises different enamel types based on their arrangement. Dentition is the highest hierarchical level, referring to the variation of *schmelzmuster* from tooth to tooth.

The change of the spatial organization of prisms from the occlusal surface to the dentin-enamel junction (DEJ) is accompanied by the gradual reduction of densities [55,56]. The main inorganic compounds of human enamel are calcium, phosphate, carbonate, magnesium, and sodium. The mass percentage of CaO and P_2_O_5_ decreases from the enamel surface to the dentin, whereas Na_2_O and MgO increases [17]. It has been shown that the highest content of Ca and P may be related to the largest values of hardness and modulus at the enamel surface. Na and Mg are mainly present in amorphous interphase (AIP) around organic proteins, so the content of Na and Mg is the highest near dentin. The presence of organic proteins and Na and Mg increases the fracture toughness of enamel, which is a function of the distance between dentin and enamel. Mg-calcium phosphate is the main phase of AIP at the grain boundary for strengthening the interfacial connections between mineral nanofibers and improving energy dissipation [57].

##### Dentin and the Dentin–Enamel Junction (DEJ)

Different from enamel, the sensitive and soft dentin forms throughout life. The 5 nm thick HAP crystallites in the collagen fibril scaffold are much less aligned in dentin. It consists of 70% HAP, 20% organics, and 10% water, with a modulus of 21.1 ± 1.3 GPa, a hardness of 0.51 ± 0.02 GPa [58], and a toughness of 1–3 MPa m^1/2^ [59]. These properties are necessary for the support of hard enamel and enable teeth to withstand lifelong masticatory stress. The distinctive cylindrical dentine tubules have the highest density near the pulp and the lowest near the enamel, and the diameter gradually decreases from the pulp to the dentin-enamel junction (DEJ) [60]. Structural overlap at the DEJ is critical to the survival of this wavy interface because it allows the stress on the tooth surface to gradually dissipate into the dentin, rather than acting as a barrier. This coupling results in a gradual change of hardness observed in this area, from ca. 4 GPa on the enamel side to ca. 1 GPa on the dentin side [60]. The gradient structure of DEJ region ensures the stress distribution and the prevention of possible microcracks.

#### 2.1.2. Other Mammal Teeth

Teeth from other primates also mostly consist of prismless enamel, radial enamel, and decussating enamel from the occlusal surface to the DEJ, although decussating enamel may be absent in small primates [51]. Teeth from other mammals such as the panda premolar [61], wild wolf fangs [62], bovine teeth [63], wild boar tusks [64], and canine posterior molars [65] are all composed of an inner dentin layer, an enamel layer, and the DEJ. This layered configuration balances the anti-abrasive properties and fracture resistance of various types of teeth. For example, the modulus and hardness of sheep molar enamel are both lower than that of human enamel, but sheep enamel is approximately 30% more resistant to wear [66].

The main biominerals of mammal teeth are HAP, but Interspecific variations in enamel crystallite compositions, sizes, and shapes are commonly observed [67]. Among human, ovine, porcine, and bovine teeth, porcine enamel has the highest content of organic matter, whereas human enamel has the lowest content. The less presence of organic matter, the larger the crystallite size, as found in human enamel [67]. Compared with the relatively small variations of crystallite sizes among these animals, twinned crystallites as thick as 500 nm were found in wild boar tusks [64], which are much thicker than the fiber of 35 nm in panda teeth [61]. Regarding the shape, although enamel crystallites are all elongated, their cross-sections are hexagonal and rhomboidal [51] or even circular [68]. An interesting case is found for colored enamel in rodent animals. AIP consists of iron compounds and amorphous calcium iron phosphate, which makes AIP harder and more acid resistant [19].

More diversity can be found in the next hierarchical level: enamel type. The first difference is evidenced in the relative orientation of interprismatic crystallites to the prisms. Although rod decussation exists in both human and pig enamel, there is a decussation plane between rod and interrod crystallites in pig enamel [69]. This decussation plane is formed due to the high angles between prismatic and interprismatic crystallites, which are arranged as sheets to partition the enamel rods. In contrast, the orientation of interprismatic crystallites in human radial enamel deviates only slightly from the long axis of prismatic crystallites. The additional decussation plane makes pig enamel less resistant to horizontal tensile stresses but enables more efficient arrest of crack propagation in multiple directions. Similar difference was found from the occlusal surface of enamel from canine and bovine [65]. The interprismatic crystallites in canine enamel are only slightly inclined to the prisms (Figure 2B); however, in bovine enamel, they are nearly perpendicular to each other (Figure 2B) [65]. This modified type of radial enamel is also found in wild boar tusks [64] but is absent in panda teeth [61,70].

**Figure 2 biomimetics-08-00042-f002:**
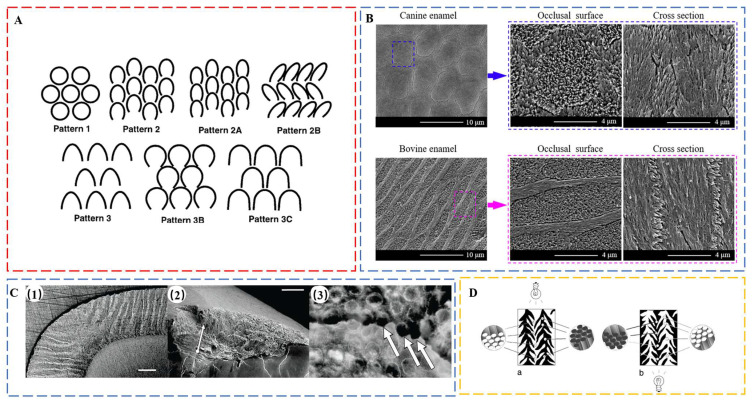
(**A**) Diagram of Boyde’s prism patterns. Pattern 1 prisms are usually small (3 to 5 µm) and have complete, roughly circular boundaries. Prisms are completely separated by interprismatic enamel and are arrayed in offset horizontal rows. Close to the outer enamel surface, the prisms of most mammals resemble Pattern 1. Boyde’s Pattern 2 and Pattern 3 defined two major classes of prisms with incomplete prism boundaries and arc-shaped prism sheaths. In both of these prism patterns, the open side of the prism is toward the cervix (root–crown junction) of the tooth. Pattern 2 prisms are small at 2 to 4 µm in diameter and are arranged in longitudinal columns from the apex to the cervix. Pattern 2 variants (Patterns 2, 2A, and 2B) differ in the amount and distribution of interprismatic enamel and angles between prisms in adjacent rows. Pattern 3 prisms are larger at 5 to 8 µm in diameter and packed in horizontally offset rows, which is similar to Pattern 1 prisms. The variants (Patterns 3, 3B, and 3C) differ in the shape of the prism sheath and the amount and distribution of interprismatic enamel. Although Boyde’s scheme has been widely used in primate enamel studies, prism patterns in many primate species are so variable that it may be more practical to distinguish only between ‘‘closed’’ (Pattern 1) and ‘‘open’’ (Patterns 2 and 3) prisms. Reproduced with permission from M.C. Maas et al. [51], Wiley. (**B**) SEM micrographs of the occlusal surfaces and axial sections of canine and bovine enamel. Reproduced with permission from H. Xiao et al. [65], Elsevier. (**C**) Sea otter enamel. (1) Light microscopy (LM) image shows high degree of decussation in sea otter enamel. HSB extend to the tooth surface. Scale bar, 0.2 mm. (2) Enamel chip; arrow indicates measurement of h. Scale bar, 0.5 mm. (3) LM at higher magnification showing the circular outlines of prisms in cross-section (each approx. 4 mm in diameter). Arrows indicate where cracks have entered prism edges. Reproduced with permission from C. Ziscovici et al. [68], the Royal Society. (**D**) Interpretation of prism direction in secondary vertical prism decussation of Crocuta, based on position of illumination and differential absorption/reflection of light by prisms. Reproduced with permission from J.M. Rensberger et al. [71], Springer Nature.

Hunter–Schreger bands (HSB) packing densities increase with loadings and are the highest in areas with concentrated stress, such as the occlusal surfaces of posterior teeth and the incisal regions of incisors [54]. The HSB packing density of otter enamel is 19.4 HSB mm^−1^, in contrast to 14 HSB mm^−1^ in human enamel, contributing to the 2.5 times higher toughness of the former [68] (Figure 2C). Considerable interspecific variations in HSB can be found in some bone-feeding carnivora, such as extant hyenas [71] (Figure 2D). Distinct from the normal undulating horizontal HSB, adjacent zigzag HSB are joined by vertical decussating prism bundles, which radiate outward from the central axis of the tooth when viewed from the transverse section. The structure resists fracture under stresses from different directions and represents an adaption to withstand remarkably high stresses during occlusion, a specialization for bone-eating.

To adapt to feeding habits, the tooth size, enamel thickness, and cuspal morphologies of mammals also vary toward optimized load-bearing abilities. Cattle need to chew for a long time, about 10 h a day, and the broad enamel certainly needs good wear and fatigue resistance. Bone-feeding carnivores, however, need much greater bite forces from teeth, which are, therefore, much sharper than cattle teeth. Studies have also shown that among primates, morphological variations may play a more important role in the different load-bearing abilities of molar teeth than materials properties such as modulus and hardness [72].

### 2.2. Aquatic Animal Teeth

#### 2.2.1. Fluorapatite Teeth

##### Shark Enameloid

Shark teeth are produced and discarded tens of thousands of times during the lifetime of a shark [22]. Distinct from human enamel, shark tooth enameloid consists mainly of fluorapatite (FAP) with a minor amount of collagen. The 5–6% content of fluoride ions in the crystal lattice ensures a better protection effect against acid [73]. Although FAP is harder than HAP, the modulus and hardness of shark teeth and human teeth is comparable both for dentin and enameloid/enamel [74]. A comparison of the tearing teeth of *Isurus oxyrinchus* with the cutting teeth of *Galeocerdo cuvier* showed that the different biological functions of shark teeth are mainly controlled by their geometry [74], a common factor for maximizing the bite forces in the teeth of piranha [75], black drum fish [76], and black carp [77]. *Isurus oxyrinchus* enameloid has FAP crystallites that are 50–80 nm thin and more than 1 μm long [78]. Underneath the shiny layer at the surface, where the crystallites are randomly oriented, there are circumferential bundles, radial bundles, and axial bundles down to the dentin (Figure 3A). The size of radial bundles decreases from the dentin–enameloid junction towards the distal layers of the enameloid, where they proceed between the circumferential bundles and project into the superficial shiny layer. Similar to *Isurus oxyrinchus* and *Galeocerdo cuvier*, most other shark teeth also have a superficial shiny layer and an inner layer consisting of crystallite bundles with changing degrees of structural organization from distal to proximal. However, there are some exceptions. The shiny layer of enameloid is absent in the teeth of *Carcharhinus plumbeus*, and in some species, the bundled layer is replaced by thick dentin [79].

In the grasping teeth of the Port Jackson shark, a graded alignment of FAP crystallites in the cusp was observed [80]. The FAP crystallites in the outer enameloid are aligned and parallel to the surface and gradually transition into a tangled organization in the inner enameloid. The graded architecture provokes a location-specific damage response for a mechanism of shape preservation. A crack-guiding effect promotes the circumferential chipping damage in the outer enameloid, whereas a crack deflection strategy slows erosion by dissipating energy in the inner enameloid.

##### Parrotfish Enameloid

FAP as the main tooth component also prevails in many other fish species, such as parrotfish. The enameloid of *Chlorurus microrhinos* is even stiffer than that of shark teeth [81], consisting of 100 nm wide, micrometer-long FAP crystals co-oriented and assembled into interwoven bundles. These fibers gradually decrease in average diameters from ca. 5 μm at the back to ca. 2 μm at the tooth tip (Figure 3B), which corresponds to the gradient in modulus and hardness. The modulus is 124 ± 8 GPa, and the hardness is 7.3 ± 0.5 GPa near the biting surface. The individual tooth exhibits a quasi-plastic contact response with a high indentation yield strength of ca. 6 GPa, resulting in extremely high abrasion resistance. The nanoscale structure and the interwoven fibers contribute to the high toughness of ca. 2.5 MPa m^1/2^.

##### Black Drum Fish Enameloid

The black drum fish (*Pogonias cromis*) enameloid is the stiffest reported so far, reaching a modulus of 126.9 ± 16.3 GPa and a hardness of 5.0 ± 1.4 GPa [76] (Figure 3C). This is attributed to the stiffening effect of Zn and F doping in apatite crystals and the preferential co-alignment of crystallographic c-axes and enameloid rods along the biting direction. In the inner enameloid region, the apatite crystals are arranged into intertwisted rods with crystallographic misorientation for increased crack resistance and toughness. The high fracture toughness of ca. 1.12 MPa m^1/2^ of the outer enameloid also promotes local yielding instead of fracture during crushing contact with mollusk shells. Similar with the teeth of parrotfish, the diameters of enameloid rods decrease from the dentin–enameloid junction to the outer surface.

##### Crayfish Mandible

An unusual, crystalline enamel-like FAP layer was found in the mandibles of the arthropod *Cherax quadricarinatus* (freshwater crayfish) [82]. Similar to most other tooth structures, crayfish teeth also have three elements: a hard covering, a soft support, and a hierarchical interface between these layers. Amorphous calcium carbonate, amorphous calcium phosphate, calcite, and fluorapatite all coexist in well-defined functional layers in proximity within the mandible.

#### 2.2.2. Self-Sharpening Teeth

##### Calcite Teeth in Sea Urchin

Self-sharpening teeth of sea urchin are made of calcite. They rely on chipping to maintain shape edges; however, the tooth grows continuously and moves out to the front from its base. The teeth of California purple sea urchin, *Strongylocentrotus purpuratus*, are made of calcite plates, fibers, and the polycrystalline matrix between them. The polycrystalline matrix comprises 10 to 20 nm nanoparticles of Mg-calcite [27,83]. There are organic layers surrounding plates and fibers that behave as the “fault lines” in the tooth structure [84]. Shedding of tooth components at these discontinuities exposes the robust central part of the tooth, “the stone”, which becomes the grinding tip (Figure 3D). The central stone part is comprised of Mg-rich polycrystalline matrix nanoparticles, and narrower fibers than those in the lateral side. These small fibers run roughly parallel to the length of the stone part [85]. An in situ wear test also confirmed that the structural geometry and chemical makeup of the tooth components are the key for maintaining sharp tips [86].

##### Iron Oxide Teeth in Chiton and Limpet

The chiton teeth are also self-sharpening, as enabled by a similar mechanism of breaking at pre-determined locations [28]. The region close to the anterior surface of the tooth cusp of *Acanthochiton rubrolineatus* has a microstructure oriented parallel to the abrasive surface with an average hardness of Hv270, whereas the underneath area has a microstructure oriented perpendicular to the abrasive surface with an average hardness of Hv490. The different abrasive property of two regions results in the faster wear down of the anterior surface than the underneath area so that the tooth keeps a sharp cutting edge under the characteristic grazing action. Chiton tooth caps from *Cryptochiton stelleri.* are three times harder than human enamel and are the hardest biominerals investigated so far (9–12 GPa) [87]. The main reason is that chiton teeth are covered in magnetite, a harder biomineral than HAP and FAP [88]. Chiton teeth also achieve impressive mechanical properties through a layered composite structure, which consists of a hard shell of organically wrapped and highly oriented nanostructured magnetite rods surrounding a soft core rich in organic iron phosphate [30] (Figure 3E). The core–shell structure, which is hard on the outside and soft on the inside, provides strong cushioning and shock absorption. The magnetite and chitin fibers support each other in different directions to make the teeth less likely to bend and break. The regular internal structure inhibits crack formation, and even when cracks do form, they spread along the gaps between the structural units, preventing the entire tooth from collapsing. Another example of iron oxide teeth is from limpet [89]. The highest recorded tensile strength (3.0 to 6.5 GPa) of limpet teeth among biological materials is attributed to a high mineral volume fraction of reinforcing goethite nanofibers with diameters below a defect-controlled critical size, which are aligned with chitin matrix.

#### 2.2.3. Transparent Teeth

##### An Invisible Weapon for Dragonfish

Although the main function of teeth is concerned with mechanical properties, the deep-sea dragonfish, *Aristostomias scintillans*, has evolved transparent teeth that are invisible under bioluminescence, showing no contrast to the surrounding water [90] (Figure 3F). The high transparency is a unique property that is the result of the nanoscale texture of teeth. The enamel-like layer is highly mineralized, consisting of amorphous/nanocrystalline HAP (ca. 20 nm grain size), whereas the dentin consists of an array of interpenetrating nanorods (ca. 5 nm in diameter) of HAP embedded within a collagen matrix. Furthermore, dragonfish teeth are also sufficiently thin (ca. 60 μm) and lack microscale features such as dentin tubules, effectively reducing Rayleigh scattering. The importance of the transparent array of hard sharp teeth may be related to camouflage. The adaptation to thriving in the aphotic zone produces a deadly invisible weapon for dragonfish.

**Figure 3 biomimetics-08-00042-f003:**
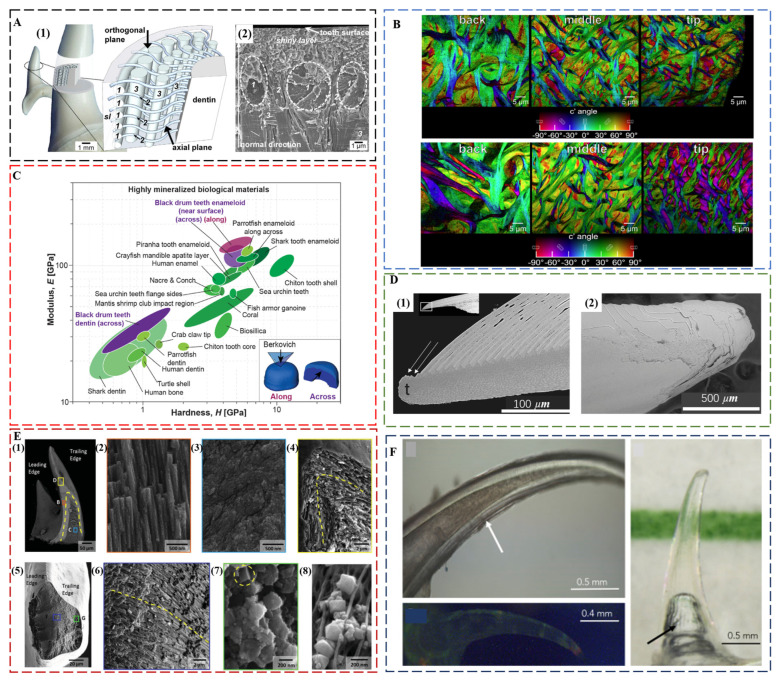
(**A**) Enameloid microstructure of teeth of *I. oxyrinchus*. (1) Schematic model depicting the locations of the analyzed tooth regions and the spatial arrangement of the fluoroapatite crystallite bundles in the enameloid. The shapes and trajectories of the bundles are simplified, and their sizes are not in scale to each other. (2) Scanning electron micrograph showing the real arrangement of differently oriented crystallite bundles and the ultrastructure of the shiny layer on a fracture plane corresponding to the axial plane of the sketch in (1). sl—shiny layer; 1—circumferential bundles; 2—radial bundles; and 3—axial bundles. Reproduced with permission from J. Enax et al. [78], Elsevier. (**B**) Polarization-dependent imaging contrast maps showing the size and orientation of fibers at the back, the middle, and the tip of the enameloid layer of the biting tooth of parrotfish. Reproduced with permission from M.A. Marcus et al. [81], American Chemical Society. (**C**) Ashby plot of modulus vs. hardness, including the properties of the black drum fish tooth enameloid and dentin measured along and across the biting direction. Reproduced with permission from Z.F. Deng et al. [76], Elsevier. (**D**) The pre-determined breaking positions in self-sharpening sea urchin teeth. (1) The convex side of the tooth tip of sea urchin, which is at the top in this longitudinal section, sheds one primary plate at a time. (2) The tip, which is seen here from the convex side, is clearly not continuously breaking but discretely shedding plates one at a time. Reproduced with permission from C.E. Killian et al. [84], Wiley. (**E**) Core–shell architecture and local microstructural features of chiton teeth. (1) Backscattered electrons (BSE) microscopy overview of a longitudinal fracture, highlighting a heavily mineralized outer shell and organic rich core. (2) Aligned and staggered nanorods in the leading edge of the shell. (3) Amorphous core region. (4) Apex of longitudinal fracture, revealing nanorod orientation and continuity around the shell of the tooth. (5) Latitudinal fracture near the tip of the tooth. (6) Center of the latitudinal fracture, demonstrating curvature of nanorods following the contour of the tooth. (7) Latitudinal fracture within the shell region, highlighting highly oriented nature of nanorods. (8) Micrograph of partially mineralized tooth showing mineral formation along alpha-chitin fibrils. Reproduced with permission from L.K. Grunenfelder et al. [30], Wiley. (**F**) Top left: close-up light microscopy, showing optical evidence of concentric layers and hollowness of the tooth. Image taken in filtered seawater with an immersive lens. Arrow points to the striations seen on the concave side. Right: Tooth imaged in seawater with color line behind to demonstrate transparency. The arrow indicates an air bubble within the hollow cavity of the tooth due to dissection from the jaw. Image captured with a polarizing filter. Bottom left: Tooth under fluorescence excited at a broadband excitation 440–490 nm and collected with a long-pass filter (>515 nm). This image shows little fluorescence. Image taken of a dry specimen in air. Reproduced with permission from A. Velasco-Hogan et al. [90], Elsevier.

## 3. Enamel-Inspired Biomimetic Materials

Tooth enamel with excellent mechanical properties establishes new design criteria for the synthesis of bioinspired and biomimetic structural materials. A brief survey of the state-of-the-art materials of this field is given here. For a comprehensive evaluation, readers can refer to recent reviews on this topic [91,92]. The seminal work of Kotov’s group used ZnO nanorods as the inorganic motif, and the interrod gaps were filled by layer-by-layer (LBL) assembly of polyallylamine (PAAm) and polyacrylic acid (PAA) [43] (Figure 4A). The multilayered columnar nanocomposites captured the key fibrous architecture of tooth enamel, combining a modulus of 39.8  ±  0.9 GPa, a hardness of 1.65  ±  0.06 GPa, and a viscoelastic figure of merit (VFOM) of 0.7 to 0.9 GPa, which surpasses traditional materials (0.6 GPa). The weight adjusted VFOM also exceeds a presumed limit of 0.8 in structural materials (Figure 4A). Following the same concept, TiO_2_ nanorods [44] and fluorapatite elongated crystallites [93] were also used to prepare abiotic tooth enamel. Distinct from LBL assembly [43,44,93], 3D printing [94] and magnetically assisted slip casting [95] allowed the synthesis of free-standing tooth replicates. A recent breakthrough was made by a bi-directional freezing process (Figure 4B). A poly-dimethylsiloxane wedge substrate induced a temperature gradient in the dispersion of amorphous ZrO_2_ (A-ZrO_2_) coated HAP nanowires with the addition of polyvinyl alcohol (PVA) [45]. This bi-directional temperature gradient directed the ice crystal growth, which in turn forced the nanowires to occupy the gaps between ice-lamellae in a parallel fashion. The multiscale engineered artificial enamel has an exceptional combination of mechanical properties: a modulus of 105.6 ± 12.1 GPa, a hardness of 5.9 ± 0.6 GPa, a flexural strength of ca. 142.9 MPa, a facture toughness of 7.4 ± 0.4 MPa m^1/2^, and a VFOM of 5.5 GPa (Figure 4B).

## 4. Outlook and Perspective

### 4.1. Hierarchical and Gradient Structures

Recent development of enamel mimetics mainly focused on the highly conserved fibrous architecture [43,44,45]. However, better mechanical performance can still be expected by imposing hierarchical design. The synthesis of comparable prismless and prismatic inorganic building units may be useful for the study of critical roles of hierarchy in affecting the mechanical properties. Various patterns of enamel prisms, including interprismatic rods and enamel types with different decussation widths and density, may deserve more attention for the replication of whole multiscale structure. The role of crystal misorientation in controlling hardness and toughness may also be considered [96,97]. Another key feature that is missing in the current enamel mimetics is the functional gradient, which is critical for mitigating stress concentration. To realize the distinct chemical (composition and density) and structural (size, shape, and orientation) gradients in columnar nanocomposites, innovative synthesis combining both physical and chemical means may be useful.

### 4.2. Multifunctionalities

The remarkable, diverse teeth discussed in this study set the basis as exciting blueprints for further development of multifunctional high-performance biomimetic materials. The intricate mouth and teeth of the sea urchin were already used as the model for a claw-like device to sample sediments on other planets, such as Mars [98] (Figure 5A). At some point, the compositional and structural features of self-sharpening teeth from sea urchins and chitons may inspire the successful synthesis of abrasion-resistant materials for tooling and machining applications. Innovative coatings for equipment and medical implants can also be envisioned. The integration of anisotropic optical and mechanical functionalities may become possible by duplicating the transparent teeth from deep-sea dragonfish.

The regeneration of teeth is an attractive property that would be highly useful in artificial replicates [22]. Even for human teeth that are diphyodont, self-healing ability was observed [99,100]. The formation process and hierarchical structure of diverse teeth can inspire the next generation of dental restorative materials, which can solve increasingly serious oral problems [101,102,103]. In a recent study, by mimicking the enamel–dentin interface, we achieved a biomimetic tooth replicate that is hard, damage tolerant, and self-healing [104] (Figure 5B). The key is the interdigitation of an enamel-like crown consisting of β-FeOOH nanocolumns with a flexible self-healing layer. Similar with the layered tooth structure, the top layer provides high modulus and hardness, whereas the bottom layer enables the energy dissipation pathway. Interestingly, upon surface fracture, the bottom layer facilitates self-healing of the top layer by allowing upward polymer diffusion to seal the damage. The biomimetic replicate is also antibacterial and has ultralow thermal diffusivity.

**Figure 5 biomimetics-08-00042-f005:**
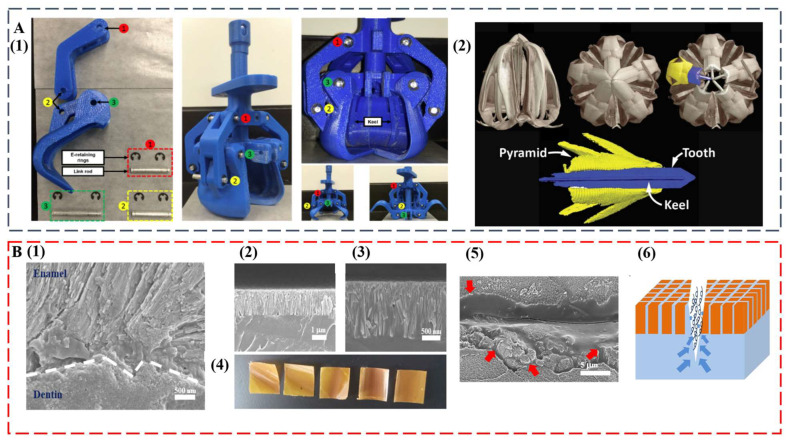
(**A**) A claw-like device inspired by sea urchin teeth. (1) Assembled 3D printed bioinspired Aristotle’s lantern parts. (2) Micro-computed tomography analysis of Aristotle’s lantern structure. Reproduced with permission from M.B. Frank et al. [98], MyJoVE Corporation. (**B**) A biomimetic tooth replicate that is hard, damage tolerant, and self-healing. (1) Human tooth enamel–dentin interface. (2 and 3) Low and high magnification cross-sectional SEM images of bilayer tooth replicate showing an interdigitated interface. (4) Optical images of bilayer tooth replicate with the nanocolumn layer on the top. (5) An SEM image of the crack after 10-min immersion in water (the red arrows indicate polymer diffusions from the bottom layer). (6) Schematic illustration of the upward polymer diffusion from the bottom layer. Reproduced with permission from Y. Yan et al. [104], Chinese Chemical Society.

### 4.3. Synthesis Challenge: Precision vs. Scalability

The synthesis of complex, versatile, and sustainable tooth-inspired materials will benefit from computer simulations, which have already shown great advantage in understanding the structure–property relationship of biomaterials [105,106] and guiding the rational materials design [107,108,109,110,111]. In the future, precise manufacture will be supported by multiscale simulation techniques and data-driven approaches [112,113] for the highly efficient replication of diverse tooth structures with optimized functions. A high precision of synthesis, however, should not compensate the scalability. Transitioning from the lab product to industrial production is always challenging for biomimetic materials [114,115]. Recent development of 3D and 4D printing with nanometer resolution is highly promising in combining the high precision and good scalability [116,117].

## Figures and Tables

**Figure 1 biomimetics-08-00042-f001:**
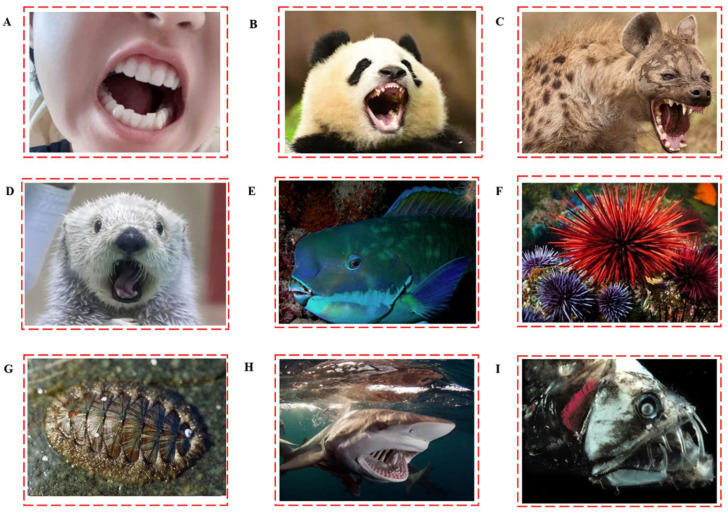
Representative animals covered in this review. (**A**) Human. (**B**) Panda. (**C**) Hyena. (**D**) Sea otter. (**E**) Parrotfish. (**F**) Sea urchin. (**G**) Chiton. (**H**) Shark. (**I**) Dragonfish. Whenever relevant, images were reproduced with permission from the website.

**Figure 4 biomimetics-08-00042-f004:**
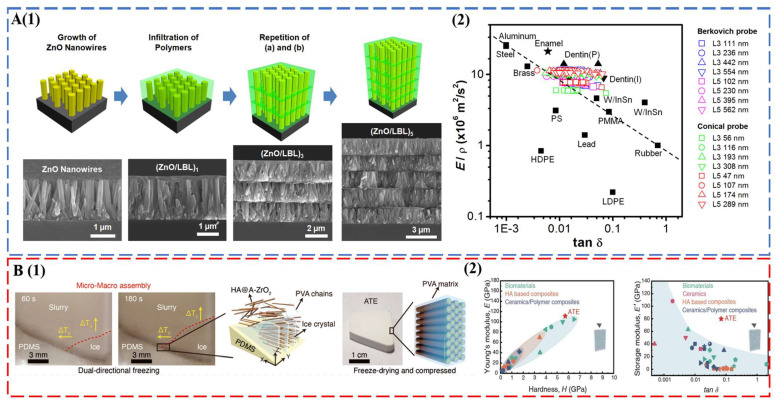
(**A**) Abiotic tooth enamel by LBL assembly. (1) Schematic illustration of the preparation and structure of columnar biomimetic composites produced by sequential LBL infiltration of ZnO nanowires with polymers. ZnO nanowires grown on an Si substrate. (2) Energy dissipation (tan*δ*) and load-bearing characteristics (*E*′/*ρ*) of (ZnO/LBL)*_n_* compared to manufactured materials and biocomposites. These figures from B. Yeom et al. [43] were kindly provided by Prof. Nicholas A. Kotov. (**B**) Multiscale engineered artificial tooth enamel (ATE) by a freezing process. (1) Schematic illustration of micro- and macro-assembly of the HA@A-ZrO_2_ nanowires coupled with PVA. (2) Storage modulus and damping coefficient of ATE compared with biomaterials, HA-based composites, ceramics, and ceramic-based composites. Reproduced with permission from H.W. Zhao et al. [45], American Association for the Advancement of Science.

## Data Availability

Data sharing is not applicable to this article as no new data were created or analyzed in this study.

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
