# Peer review of "Tooth Diversity Underpins Future Biomimetic Replications"

_biomimetics, 2023, doi:10.3390/biomimetics8010042_

Round 1

Reviewer 1 Report

This is a timely review that provides a comprehensive summary about tooth diversity and recent achievements. The quality of this review is generally in-line with biomimetics, until the following comments were addressed:

(1)  Human is a typical mammal, why does the author describe human teeth alone? Although researchers have studied the microstructure and mechanical properties of human teeth thoroughly, the author has not a clear understanding of what makes human teeth special from those of other mammals.

(2)  The references in the legend cited by the author do not specify copyright information.

(3)  The format of the scale bar in the figure should be uniform, such as in Figure 3D. Please check all the figures in detail.

(4)  The proportions of some pictures are compressed resulting in an aesthetic impact, such as in Figure 4. Please check all the figures in detail.

(5)  The formation process and hierarchical structure of diverse teeth can inspire the next generation of dental restorative materials, which can solve increasingly serious oral problems (Adv. Mater. 2020, 32, 2002080; Adv. Mater. 2020, 32, 1907067; Sci. Adv. 2019, 5, eaaw9569). Please supplement the multifunctional section.

Author Response

Reviewer 1.

This is a timely review that provides a comprehensive summary about tooth diversity and recent achievements. The quality of this review is generally in-line with biomimetics, until the following comments were addressed:

  • Human is a typical mammal, why does the author describe human teeth alone? Although researchers have studied the microstructure and mechanical properties of human teeth thoroughly, the author has not a clear understanding of what makes human teeth special from those of other mammals.

Our response: We agree with the reviewer that there is not enough reason that human teeth should be listed separately. So, in the revised version, we changed the subtitles to include human teeth and other mammal teeth under the section of mammal teeth.

  • The references in the legend cited by the author do not specify copyright information.

Our response: The copyright information was added in the revised version.

  • The format of the scale bar in the figure should be uniform, such as in Figure 3D. Please check all the figures in detail.

Our response: We carefully checked the scale bar in all the figures and the scale bar in Figure 3D was modified.

  • The proportions of some pictures are compressed resulting in an aesthetic impact, such as in Figure 4. Please check all the figures in detail.

Our response: We carefully checked all the pictures and besides Figure 4, Figure 5 was also corrected.

(5)  The formation process and hierarchical structure of diverse teeth can inspire the next generation of dental restorative materials, which can solve increasingly serious oral problems (Adv. Mater. 2020, 32, 2002080; Adv. Mater. 2020, 32, 1907067; Sci. Adv. 2019, 5, eaaw9569). Please supplement the multifunctional section.

Our response: We are sorry for the omission of such an important topic regarding the regeneration of teeth using biomimetic approach. The recommended sentence was included in the new version.

Reviewer 2 Report

This work summarized the nanostructures of the teeth from various biomaterials. The differences between the systems are well described, followed by introductions to artificial nanocomposites and perspectives. I think this work will be useful for the researchers and general audiences who want to investigate the structural and mechanical aspects of the teeth and similar shaped nanocomposites. Therefore I recommend this article after minor revisions. Some figure images such as Figure 4A and B are somewhat elongated in vertical direction with mismatched aspect ratios. Also Figure 3C needs higher resolution than now.

Author Response

This work summarized the nanostructures of the teeth from various biomaterials. The differences between the systems are well described, followed by introductions to artificial nanocomposites and perspectives. I think this work will be useful for the researchers and general audiences who want to investigate the structural and mechanical aspects of the teeth and similar shaped nanocomposites. Therefore I recommend this article after minor revisions. Some figure images such as Figure 4A and B are somewhat elongated in vertical direction with mismatched aspect ratios. Also Figure 3C needs higher resolution than now.

Our response: The issues regarding the elongated images were corrected and Figure 3C with higher resolution was used in the revised version.

Reviewer 3 Report

Thank you for allowing me to review this scientific article, the purpose of which was to the purpose of which is to highlight the fact that the diversity of teeth is the basis of future biomimetic replications.

The topic of the paper is very interesting and very well argued. I don't have many observations to make. First of all, figure 2 should be moved higher, after 1. Human teeth. Enamel (line 103).

I would recommend checking the English language once more.

Author Response

Thank you for allowing me to review this scientific article, the purpose of which was to the purpose of which is to highlight the fact that the diversity of teeth is the basis of future biomimetic replications.

The topic of the paper is very interesting and very well argued. I don't have many observations to make. First of all, figure 2 should be moved higher, after 1. Human teeth. Enamel (line 103).

I would recommend checking the English language once more.

Our response: Thank you for reading our review paper. The new version has repositioned all the figures. We also carefully check the English writing again.

Reviewer 4 Report

The topic discussed in the manuscript was very interesting and novel, which is very informative in the field of dentistry

Author Response

The topic discussed in the manuscript was very interesting and novel, which is very informative in the field of dentistry.

Our response: We appreciate your high opinion on our review paper.